# Dynamic control of adipose tissue development and adult tissue homeostasis by platelet-derived growth factor receptor alpha

Sunhye Shin[1†], Yiyu Pang[1†], Jooman Park[1], Lifeng Liu[1], Brandon E Lukas[1], Seung Hyeon Kim[2], Ki-Wook Kim[2], Pingwen Xu[3], Daniel C Berry[4], Yuwei Jiang[1]*

[1]Department of Physiology and Biophysics, College of Medicine, The University of Illinois, Chicago, United States; [2]Department of Pharmacology, College of Medicine, The University of Illinois, Chicago, United States; [3]Division of Endocrinology, Department of Medicine, The University of Illinois at Chicago, Chicago, United States; [4]Division of Nutritional Sciences, Cornell University, Ithaca, United States

**Abstract** Adipocytes arise from distinct progenitor populations during developmental and adult stages but little is known about how developmental progenitors differ from adult progenitors. Here, we investigate the role of platelet-derived growth factor receptor alpha (PDGFRα) in the divergent regulation of the two different adipose progenitor cells (APCs). Using in vivo adipose lineage tracking and deletion mouse models, we found that developmental PDGFRα+ cells are adipogenic and differentiated into mature adipocytes, and the deletion of *Pdgfra* in developmental adipose lineage disrupted white adipose tissue (WAT) formation. Interestingly, adult PDGFRα+ cells do not significantly contribute to adult adipogenesis, and deleting *Pdgfra* in adult adipose lineage did not affect WAT homeostasis. Mechanistically, embryonic APCs require PDGFRα for fate maintenance, and without PDGFRα, they underwent fate change from adipogenic to fibrotic lineage. Collectively, our findings indicate that PDGFRα+ cells and *Pdgfra* gene itself are differentially required for WAT development and adult WAT homeostasis.

*For correspondence:
yuweij@uic.edu

[†]These authors contributed equally to this work

Competing interests: The authors declare that no competing interests exist.

## Introduction

White adipose tissue (WAT) is a dynamic endocrine organ that controls important physiological processes and mediates various metabolic responses (*Kershaw and Flier, 2004*; *Rosen and Spiegelman, 2006*; *Rosen and Spiegelman, 2014*; *Spiegelman and Flier, 2001*; *Trayhurn and Beattie, 2001*). However, the development of WAT is not well understood. Adipocytes are constantly replenished with new adipocytes derived from the stem cell pool, a process named adipogenesis (*Cawthorn et al., 2012*; *Sebo and Rodeheffer, 2019*). In young adult mice, the rate of adipogenesis has been estimated at 10–15% per month (*Rigamonti et al., 2011*), and retrospective human studies also indicate a high turnover rate (*Spalding et al., 2008*). Under homeostatic conditions, the process is relatively constant, but it is sensitive to pharmacologic, physiologic, and dietary stimuli. For instance, adipose tissues can expand from 2–3% to 60–70% of body weight in response to a positive energy balance through both hyperplasia and hypertrophy (*Ginsberg-Fellner, 1981*; *Hirsch and Batchelor, 1976*; *Hirsch and Knittle, 1970*; *Jo et al., 2009*; *Knittle et al., 1979*). Notably, the thiazolidinedione (TZD) class of diabetes treatments increases de novo adipogenesis by stimulating stem cell compartment self-renew and proliferation (*Tang et al., 2011*). Both childhood and adult obesity are caused by uncontrolled expansion of WAT and excessive lipid accumulation, which elevate the risk of metabolic disorders (*Berry et al., 2016a*; *Hajer et al., 2008*; *Jiang et al., 2012*;

*Smorlesi et al., 2012*). However, the underlying differences between these two types of obesity are not clear yet. Therefore, there is a clear clinical need to investigate how WAT is developed, maintained, and expanded during developmental and adult stages.

Recently, multiple genetic fate-mapping and lineage-marking studies have been conducted to understand when and where adipose progenitor cells (APCs), which are capable to proliferate and differentiate into new adipocytes, are specified (*Cattaneo et al., 2020*; *Jiang et al., 2014*; *Lee et al., 2013*; *Sanchez-Gurmaches and Guertin, 2014*; *Sanchez-Gurmaches et al., 2015*; *Sebo et al., 2018*; *Tang et al., 2008*; *Tran et al., 2012*; *Vishvanath et al., 2016*; *Wang et al., 2013*). For example, it was reported that adult adipocytes, but not developmental adipocytes, are differentiated from perivascular smooth muscle actin (SMA, encoded by *Acta2* gene) mural cell source to reside along the blood vessel walls within WAT (*Jiang et al., 2014*). The following study identified that platelet-derived growth factor receptor beta (PDGFRβ) mediates the interaction and communication between adult SMA+ APC and niche (*Jiang et al., 2017b*). Lineage tracing studies reveal that adipose mural PDGFRβ+ cells do not contribute to adult homeostasis but contribute to adipose remodeling in obese or cold exposed adult mice (*Vishvanath et al., 2016*). These findings reveal that adipocytes arise from diverse lineages and that there are at least two different adipose progenitor populations, including developmental progenitors used for adipose tissue organogenesis and adult progenitors used for adipose tissue homeostasis. However, the origin and identity of the developmental progenitors remain to be determined. Also, it is not clear whether developmental and adult progenitors utilize different regulatory mechanisms to give rise to functionally different adipocytes. Recent studies suggest that, even within a single adipose depot, there appear to be multiple subpopulations of adipocytes (*Lee et al., 2019*).

Platelet-derived growth factor receptor alpha (PDGFRα) is a membrane-bound tyrosine kinase receptor expressed in perivascular stromal cells within a variety of tissues. PDGFRα has been commonly used as a cell surface marker for adipose progenitor identification, and multiple studies have reported that PDGFRα+ cells generate adipocytes in response to adipogenic stimuli (*Berry and Rodeheffer, 2013*; *Cattaneo et al., 2020*; *Joe et al., 2010*; *Lee et al., 2012*; *Lee et al., 2012*; *Rivera-Gonzalez et al., 2016*). For example, using $Pdgfra^{Cre}$; $Rosa26R^{mT/mG}$ mice, PDGFRα marks adipocytes (*Berry and Rodeheffer, 2013*). Also, WAT-resident PDGFRα+ cells can develop into brown-like adipocytes in response to β3-adrenergic agonist or white adipocytes in response to high-fat diet feeding (*Lee et al., 2012*). Recent studies have shown that there are two subsets of PDGFRα+ cells in adipose tissues delineated by CD9 expression. Whereas CD9-high PDGFRα+ cells are pro-fibrogenic and drive adipose tissue fibrosis, CD9-low PDGFRα+ cells are pro-adipogenic and make adipocytes (*Marcelin et al., 2017*). In addition, increased PDGFRα activity drives adipose tissue fibrosis during both adult homeostasis and adipose tissue organogenesis (*Iwayama et al., 2015*; *Sun et al., 2017*). However, due to the complexity and nonspecificity of the mouse lines, our understanding of the role of PDGFRα+ cells in vivo has been limited. Further clarification of PDGFRα+ cell fate by lineage tracing studies at different time points is still needed. In addition, loss-of-function models generated in the developmental or adult adipose lineage are required to definitively determine the physiological functions of PDGFRα in different stages.

In this study, we aimed to understand the role of PDGFRα+ cells and the gene itself in different stages of adipose tissue (postnatal development and adult maintenance of WAT) using in vivo adipose lineage tracking and gene deletion systems. We found that PDGFRα+ cells are a progenitor source for postnatal WAT development but not adult WAT homeostasis. Consistently, *Pdgfra* expression in APCs is not essential for adult WAT homeostasis but required for postnatal WAT development. The deletion of *Pdgfra* in adult APCs did not disrupt adult WAT maintenance and cold-induced beige adipocyte formation. However, the deletion of *Pdgfra* in developmental APCs led to a significant fat reduction associated with smaller fat depots. Mechanistically, embryonic PDGFRα-deficient APCs were unable to differentiate into mature adipocytes and underwent fate change from adipogenic to fibrotic lineage. Together, our findings unraveled a dynamic requirement for PDGFRα+ cells and the *Pdgfra* gene itself in controlling WAT development and WAT homeostasis.

## Results

### Developmental adipocytes derive from a PDGFRα+ cell source

Our previous work demonstrated that adult but not developmental adipocytes emanate from a vascular smooth cell expressing smooth muscle actin (SMA) and other mural markers (*Jiang et al., 2017b*; *Jiang et al., 2014*). However, the specific origins of developmental APCs remain unknown. We proposed to test the possibility of using PDGFRα as a fate marker for the developmental APCs. PDGFRα is a membrane-bound tyrosine kinase receptor that has been used as a cell surface marker for adipose progenitor identification. Moreover, multiple studies using several *Pdgfra* genetic tools have shown that PDGFRα+ cells can mark the adipose lineage and generate adipocytes (*Berry and Rodeheffer, 2013*; *Lee et al., 2012*). We hypothesized that PDGFRα+ cells mark the developmental APCs but not adult APCs, and contribute to adipose tissue organogenesis and development.

To investigate the fate mapping of PDGFRα+ cells, we marked and monitored PDGFRα+ cells using $Pdgfra^{Cre-ERT2}$; $Rosa26R^{RFP}$ (PDGFRα-RFP) mice. Using this model, we previously reported $Pdgfra^{Cre-ERT2}$ dependent RFP reporter expression faithfully labels adipose tissue-resident PDGFRα+ cells and their descendants (*Berry et al., 2016b*; *Lee et al., 2012*). To avoid potential off-target effects of tamoxifen (*Ye et al., 2015*) and to induce *Pdgfra*-dependent recombination, we provided one dose of tamoxifen at P10 (adipose tissue organogenesis) or P60 (established depots for adipocyte turnover and maintenance), and examined reporter expression at pulse (P13 or P63; pulse) or after a 2-month chase (P60 or P120). Both whole-mount staining and immunohistochemistry (IHC) studies indicated that, at pulse (P13 or P63), PDGFRα-dependent RFP expression was restricted to the perivasculature in subcutaneous inguinal WAT (IGW), perigonadal WAT (PGW), and brown adipose tissue (BAT). As expected, mature adipocytes were not labeled as previously reported (*Berry et al., 2016b*; *Berry and Rodeheffer, 2013*; *Lee et al., 2012*; *Figure 1—figure supplement 1*). However, P10 to P60 chase revealed the elaboration of RFP expression into adipocytes in both male and female IGW and BAT but not in PGW suggesting the creation of new adipocytes from a PDGFRα+ source (*Figure 1A*). Of note, there was strong RFP+ labeling in the male epididymis (*Figure 1A*). Surprisingly, during the P60 to P120 chase, we observed minimal RFP-adipocyte labeling in IGW, PGW, and BAT depots (connected with interscapular WAT) based on whole-mount images (*Figure 1B*). Interestingly, we did not observe fate mapping differences between male and female adipose depots (*Figure 1A,B*).

To further deduce the contribution of PDGFRα+ lineage during WAT development and maintenance, we quantified RFP+ cells in adipose tissue sections from PDGFRα-RFP mice (RFP marks PDGFRα+ lineage and their descendants) injected with tamoxifen at P10 or P60 and perfused at P60 or P120. Consistent with the results from whole-mount imaging, there were RFP+ adipocytes in IGW and BAT depots but not PGW depots from P10-P60 chow-fed male mice (IGW 20–30%; PGW 0%; BAT 30–35%) (*Figure 1C*). In contrast to developmental labeling, we observed significantly less RFP + (roughly 2%) adipocyte labeling of WAT and BAT depots from male mice. Rather our IHC studies demonstrated the presence of PDGFRα-RFP+ cells residing in perivascular positions, similar to pulse (*Figure 1D*). In a separate study, PDGFRα-reporter mice were administered tamoxifen at P60 and chased to P180. Again, we observed very few adipocytes labeled by PDGFRα-RFP+ cells rather these cells appeared to be restricted to the vasculature of both WAT and BAT (*Figure 1—figure supplement 1*). Together, it appears, under our conditions, that PDGFRα+ cells give rise to developmental adipocytes but not a major APCs for adult adipocytes.

### Developmental, but not adult, PDGFRα+ cells are a cellular origin of adipogenesis associated with high-fat diet and TZD feeding

It has been reported that PDGFRα+ cells contribute to adipose tissue expansion in response to high-fat diet (HFD) (*Lee et al., 2012*). To test if the fate-mapping potential of PDGFRα-RFP+ cells changes in response to HFD challenge, we fed tamoxifen-induced PDGFRα-RFP reporter mice from P10-P60 or P60-P120 with chow diet or HFD (60% of calories from fat). To our surprise but in agreement with our fate-mapping studies above, we found very few PDGFRα-RFP+ generated adipocytes during the P60-120 HFD challenge (*Figure 1B,D*). By contrast, developmental HFD fate mapping demonstrated a strong correspondence between RFP reporter expression and adipocytes labeling in IGW and BAT but not PGW. Quantification indicated that ~90% of IGW and ~95% of brown

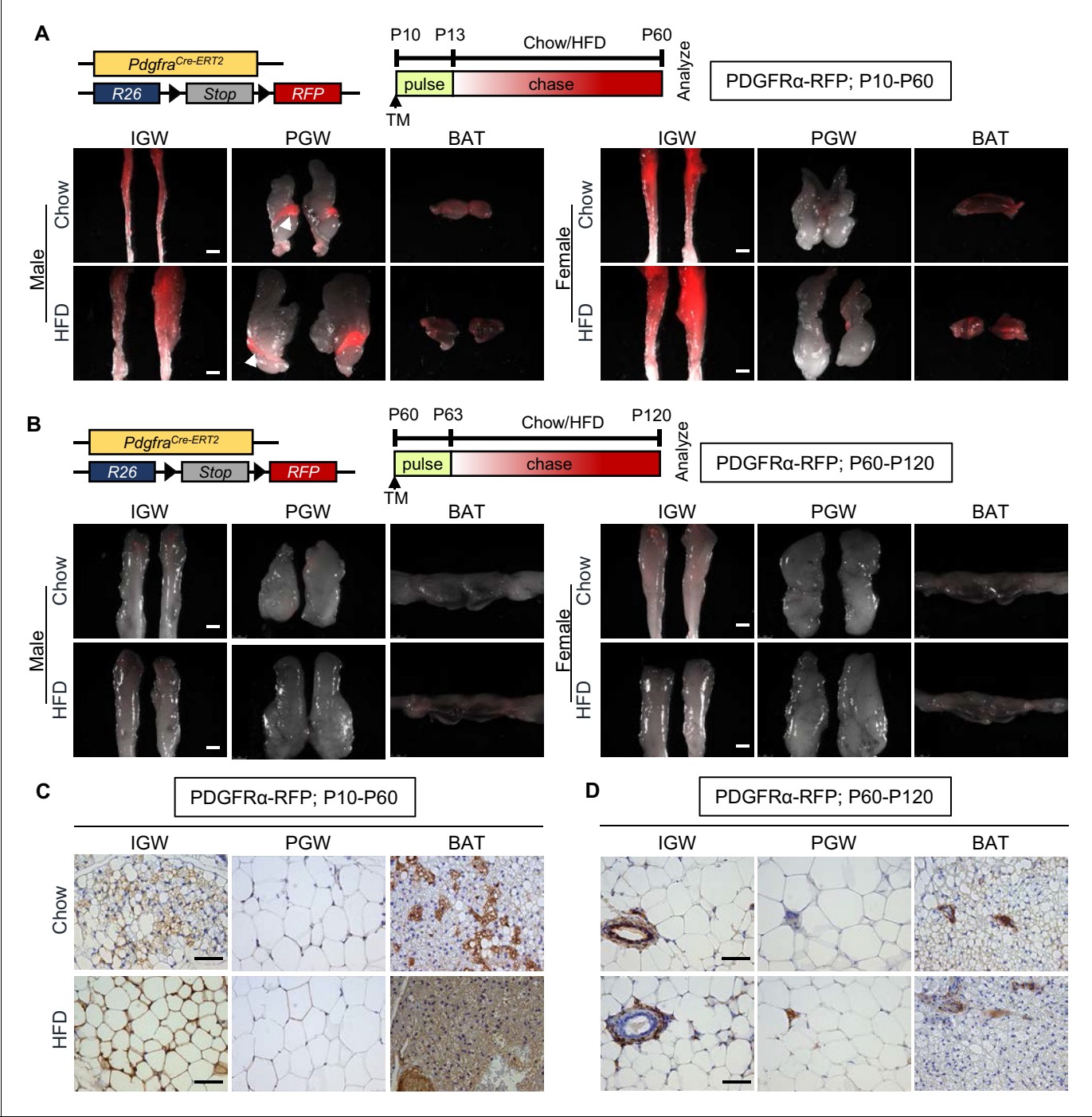

**Figure 1.** Developmental, but not adult, adipocytes derive from a PDGFRα+ cell source. (**A–B**) *Pdgfra*$^{Cre-ERT2}$*; Rosa26R*$^{RFP}$ (PDGFRα-RFP) mice were administered tamoxifen (TM) (**A**) at postnatal day 10 (**P10**) and fed chow or HFD until P60 or (**B**) at P60 and fed chow or HFD until P120. IGW, PGW, and BATs were examined for direct RFP fluorescence either at (**A**) P60 or (**B**) P120 (chase). White arrowheads indicate the epididymis labeling. Scale = 100 μm. (**C–D**) RFP staining of IGW, PGW, and BATs from above P10-P60 and P60-P120 mice using immunohistochemistry (IHC). Scale = 200 μm. The online version of this article includes the following figure supplement(s) for figure 1:

**Figure supplement 1.** *Pdgfra*-dependent RFP expression at pulse, P60-P180, and P60-P120 TZD pulse-chase.

adipocytes were RFP+ whereas only ~5% PGW adipocytes were labeled (*Figure 1A,C*; *Figure 1—figure supplement 1*). The low labeling of PGW could reflect the developmental specification of this depot, as this depot has been shown to be specified beyond P10. Together, our fate-mapping data suggest that P10 but not P60 labeled PDGFRα+ stromal vascular (SV) cells are adipogenic. HFD-fed mice utilize only the developmental but not adult labeled PDGFRα+ cells as a cellular source for adipose tissue expansion.

Peroxisome proliferator-activated receptor gamma (PPARγ) is a master regulator of adipogenesis (*Farmer, 2006*; *Lehrke and Lazar, 2005*). PPARγ agonists such as rosiglitazone (Rosi), a thiazolidinedione (TZD), have been reported to trigger the formation of new adipocytes from an adult adipose stem/progenitor compartment (*Crossno et al., 2006*; *Tang et al., 2011*). To test whether adult PDGFRα+ cells can acquire adipogenic potential when exposed to PPARγ agonists, we administered Rosi to PDGFRα-RFP+ mice for 8 weeks (*Figure 1—figure supplement 1*). We observed that the PDGFRα-dependent RFP expression remained restricted at the vasculature, and there were rare RFP + labeled adipocytes, based upon lipidTox and perilipin staining of IGW sections (*Figure 1—figure supplement 1*). These data suggest that in response to TZDs administration, PDGFRα+ cells may not represent a major progenitor cell population for new adipocytes.

## Developmental and adult PDGFRα+ cells have distinct molecular and functional signatures

To examine the in vitro adipogenic potential of P10 and P60 PDGFRα+ cells, we isolated total SV cells from IGW depots from tamoxifen-pulsed PDGFRα-RFP mice (P13 or P63; pulse). This fraction contains both RFP+ and RFP negative (RFP-) cells. Cells were cultured for 7 days in white adipogenic conditions. Consistent with our fate-mapping data, the SV cells from P13 mice produced RFP labeled mature adipocytes (>75% of total adipocytes are RFP+) (*Figure 2A*). By contrast, the SV cell cultures from P63 PDGFRα-RFP mice generate very few RFP+ adipocytes (<5% of total adipocytes are RFP+) and PDGFRα-RFP+ cells retained their fibroblast morphology (*Figure 2B*). Using fluorescence-activated cell sorting (FACS), we isolated P10 tamoxifen-pulsed PDGFRα-RFP SV cells into RFP+ and RFP- cells and subsequently cultured them in adipogenic media for 7 days. Cultures containing FACS-isolated P10 RFP+ had many RFP+ adipocytes and were overall more adipogenic compared to P10 RFP- cells as assessed by lipid content and adipocyte gene expression (*Figure 2C,D*). These data are consistent with our in vivo lineage-tracing data, indicating that developmental PDGFRα+ cells are adipogenic.

Our in vivo lineage-tracing data and in vitro primary cell culture data indicated that WAT organogenesis requires PDGFRα+ cells while adult WAT homeostasis utilized a different APC source. We next investigated whether the molecular basis of P10 and P60 PDGFRα+ cells were distinct. We FACS-sorted P10 and P60 tamoxifen-pulsed PDGFRα-RFP SV cells into RFP+ and RFP- cells (*Figure 2E*). FACS-isolated P10 PDGFRα+ cells had significantly higher levels of preadipogenic markers (*Pparg*, *Pref1*, *Zfp423*) compared to P60 PDGFRα+ cells (*Figure 2E*). By contrast, levels of fibroblast markers (*Col1a1* and *Col3a1*) and *Cd24* (a proposed APC marker) displayed no significant difference between P10 and P60 PDGFRα+ cells (*Figure 2E*). We also did not observe any differences in the expression of mature adipocyte markers (*Fabp4*, *Plin1*, *Adipoq*) and endothelial markers (*Cd31*, *VE-cadherin*) (data not shown).

To further assess PDGFRα's contribution to the APC lineage, we combined the PDGFRα-RFP reporter mouse model with the doxycycline suppressible adipose lineage track system, AdipoTrak (*Pparg*^tTA; TRE-*Cre*; TRE-*H2B-GFP*) (*Jiang et al., 2017b*; *Tang et al., 2008*). AdipoTrak labeled cells are necessary for WAT formation and homeostasis and mark the entire adipose lineage (stem-to-adipocyte) (*Jiang et al., 2017b*; *Tang et al., 2008*). This dual model will allow for spatiotemporal lineage identification and overlap between PDGFRα-RFP+ cells and AdipoTrak-GFP+ cells (*Figure 2—figure supplement 1*). Dual reporter mice were tamoxifen-induced at P10 or P60 and SV cells from WATs were isolated 3 days later. Flow cytometric quantification, using RFP as a surrogate for PDGFRα and GFP for PPARγ, identified strong correspondence between RFP and GFP in P10 samples (IGW 15.36%; PGW 8.25%; BAT 6.63%), but not at P60 (*Figure 2F*; *Figure 2—figure supplement 1*). Together, these results indicate that P10 but not P60 PDGFRα+ cells express adipose progenitor markers, which might account for their differences in adipogenic capabilities.

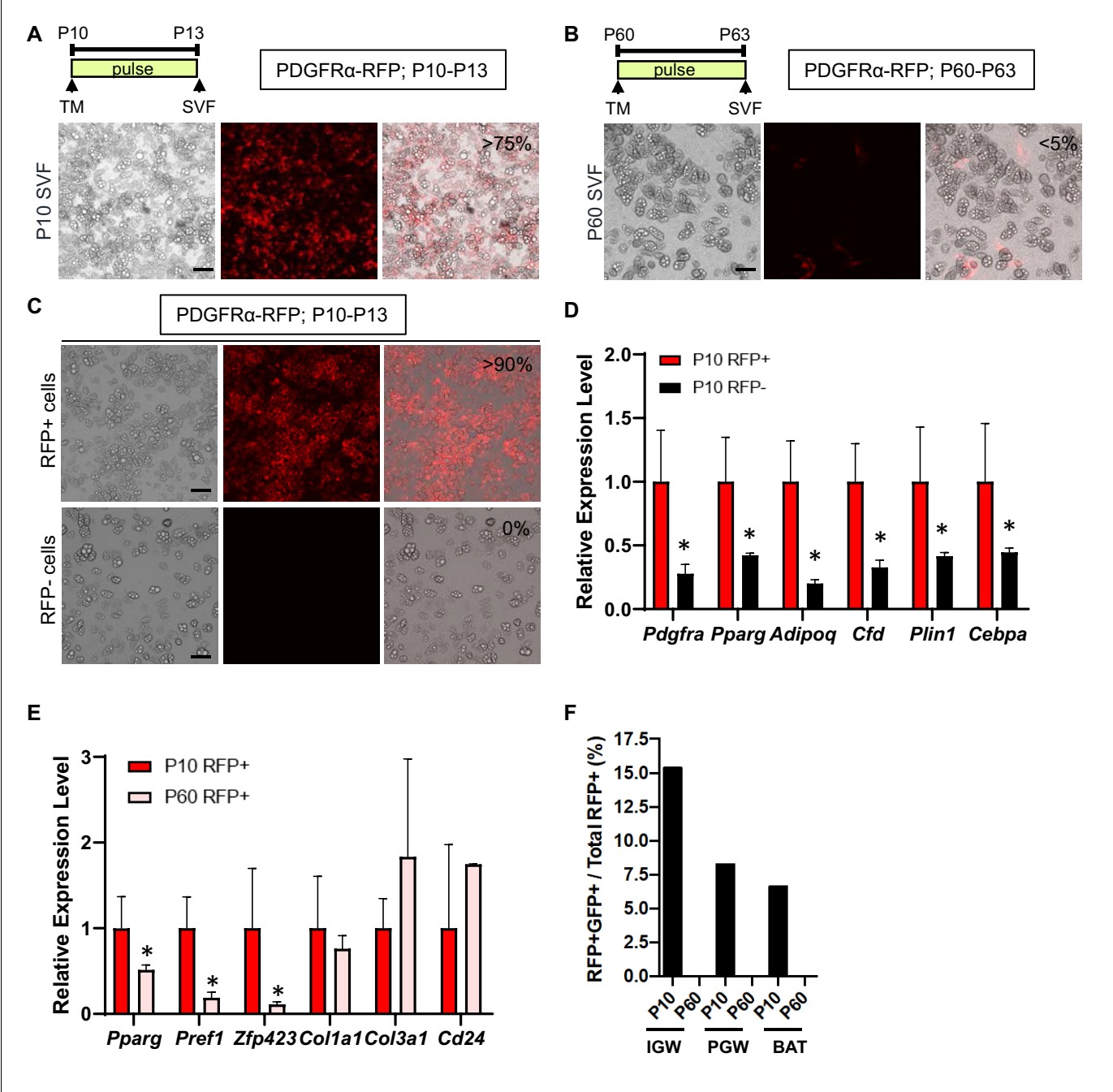

**Figure 2.** Developmental, but not adult, PDGFRα+ cells are adipogenic. (A–B) *Pdgfra*[Cre-ERT2]; *Rosa26R*[RFP] (PDGFRα-RFP) male mice were administered TM at P10 (A) or P60 (B). Stromal vascular fraction of cells (SVF) were isolated from IGW of the mice after 3 days and cultured. The numbers indicate the percentage of the RFP+ labeled adipocytes. Scale = 100 μm. (C) TM-induced PDGFRα-RFP male mice either at P10 or P60 were fed chow or HFD until P120. SVF were isolated from IGW of the mice at P120, cultured, and examined for direct RFP fluorescence. Scale = 100 μm. (D) The real-time q-PCR analysis of adipogenic markers from cells described in C. *p<0.05 P10 RFP- compared to P10 RFP+ cells. (E) Gene expression levels of P10 and P60 RFP + cells in SVF isolated from pooled IGW (n = 10). *p<0.05 P60 RFP+ compared to P10 RFP+ cells. Data are expressed as mean ± SEM. (F) *Pparg*[tTA]-*H2BGFP*; PDGFRα-RFP male mice were administered TM at P10 or P60. SVF were isolated from the pooled IGW, PGW, and BAT depots (n = 8) after 3 days and sorted using RFP signal. GFP+RFP+ cells were quantified.

The online version of this article includes the following figure supplement(s) for figure 2:

**Figure supplement 1.** Developmental, but not adult, PDGFRα+ cells overlap with PPARγ+ cells.

## Developmental PDGFRα+ cells contribute to postnatal but not adult WAT development

To test whether these P10 labeled PDGFRα-RFP+ cells still maintain adipogenic potential in the adult stage, we performed fate-mapping tests from P10-P120. WAT whole-mount imaging showed that PDGFRα+ SV cells labeled at P10 made adipocytes which could still be observed at P120 in IGW under both chow- and HFD-fed conditions (*Figure 3A*; *Figure 3—figure supplement 1*). We quantified the number of PDGFRα-RFP+ adipocytes between P10-P120 fate mapping with our P10-P60 fate mapping studies. Interestingly, we found that the percentage of RFP+ adipocyte labeling from P10-P120 was either maintained or significantly reduced compared to RFP- adipocyte labeling from P10-P60 (Chow P10-P120: IGW ~20%; PGW ~20%; BAT ~5%, HFD P10-P120: IGW ~5%; PGW ~ 10%; BAT ~10%) (*Figure 3—figure supplement 1*). Thus, our fate-mapping data suggest that postnatal P10 PDGFRα+ cells do not continue to contribute to adult WAT homeostasis or HFD-induced expansion.

To validate the notion that adult PDGFRα+ cells do not contribute to adult WAT homeostasis, we revisited the constitutive *Pdgfra*$^{Cre}$ mouse model and combined it with *Rosa26R*$^{RFP}$ reporter. Lineage marking analysis demonstrated that different labeling results in IGW depots at 2-month-old and 6–month-old mice. We found nearly all 2-month-old IGW mature adipocytes (95–100%) were labeled with RFP. Yet, at 6 months, we found minimal RFP-adipocyte marking (10–15%) (*Figure 3B*). These data suggest that adipocytes generated in the adult homeostatic phase were derived from a PDGFRα-independent source. To further confirm this, we generated a deletion model, in which PPARγ, the master regulator of adipogenesis, was constitutively deleted in PDGFRα+ cells, to block adipocyte differentiation. We observed that there was severe disruption of IGW development at 2-month-old mice, revealing the importance of PDGFRα+ cells for adipose tissue development. However, 6-month-old mice showed recovered IGW tissue size with normal adipocyte number (*Figure 3C*). These data support the possibility that developmental PDGFRα+ cells are used for WAT development, but adult WAT maintenance does not utilize PDGFRα+ cells as a progenitor source.

To further evaluate the necessity of the PDGFRα+ cells in a cell-autonomous manner, we combined the *Pparg*$^{fl/fl}$ conditional mouse model with the tamoxifen-inducible *Pdgfra*$^{Cre-ERT2}$. This model will provide a spatiotemporal deletion of *Pparg* to test the necessity of PDGFRα+ cells to generate new white adipocytes. At P60, we isolated SV cells from un-induced *Pdgfra*$^{Cre-ERT2}$; *Pparg*$^{fl/fl}$ (PDGFRα-PPARγ-KO) mice. We then cultured the cells in adipogenic media containing either vehicle or 4-OH-tamoxifen (2 uM/mL). Consistent with the in vivo lineage tracing data, SV cells from control and PDGFRα-PPARγ-KO mice that received 4-OH-tamoxifen underwent adipogenesis similarly as indicated by Oil Red O staining and adipocyte marker expression (*Figure 3—figure supplement 1*). These results support the notion that adult labeled PDGFRα+ cells are not an essential cellular source for adipogenesis.

## PDGFRα in adult SMA+ APCs is not required for adult white or beige adipogenesis under physiological conditions

Our in vivo and in vitro data show that PDGFRα+ cells do not contribute to adult WAT homeostasis. SMA+ cells were reported as adult adipose progenitor cells required for adult WAT homeostasis and turnover (*Jiang et al., 2014*). Further, this study showed that some SMA+ cells express PDGFRα. Flow analysis indicated that about half of the RFP labeled SMA+ cells expressed PDGFRα, and this was confirmed by quantitative PCR analysis, showing a ~10-fold enrichment of *Pdgfra* mRNA expression in RFP labeled SMA+ cells. Thus, we hypothesized that PDGFRα in SMA+ cells could potentially regulate APC function and differentiation. To test this notion, we combined the *Pdgfra*$^{fl/fl}$ conditional mouse model with APC lineage tracking and deletion tool, *Acta2*$^{Cre-ERT2}$; *Rosa26*$^{RFP}$ to create *Acta2*$^{Cre-ERT2}$; *Pdgfra*$^{fl/fl}$ (SMA-PDGFRα-KO; *Figure 4A,B*). At P60 mice were administered one dose of tamoxifen for 2 consecutive days and mice were analyzed 30 days later. Consistent with the PDGFRα fate-mapping studies, at P90, we observed no physiological difference between control and SMA-PDGFRα-KO mutant mice under chow diet feeding. The control and mutant mice had similar body weight (*Figure 4C*), fat mass (*Figure 4—figure supplement 1*), food intake (*Figure 4—figure supplement 1*), adipose tissue weights (*Figure 4D*), and glucose clearance (*Figure 4E*). As expected, SMA-PDGFRα-KO mutant BAT had reduced expresson of *Pdgfra*

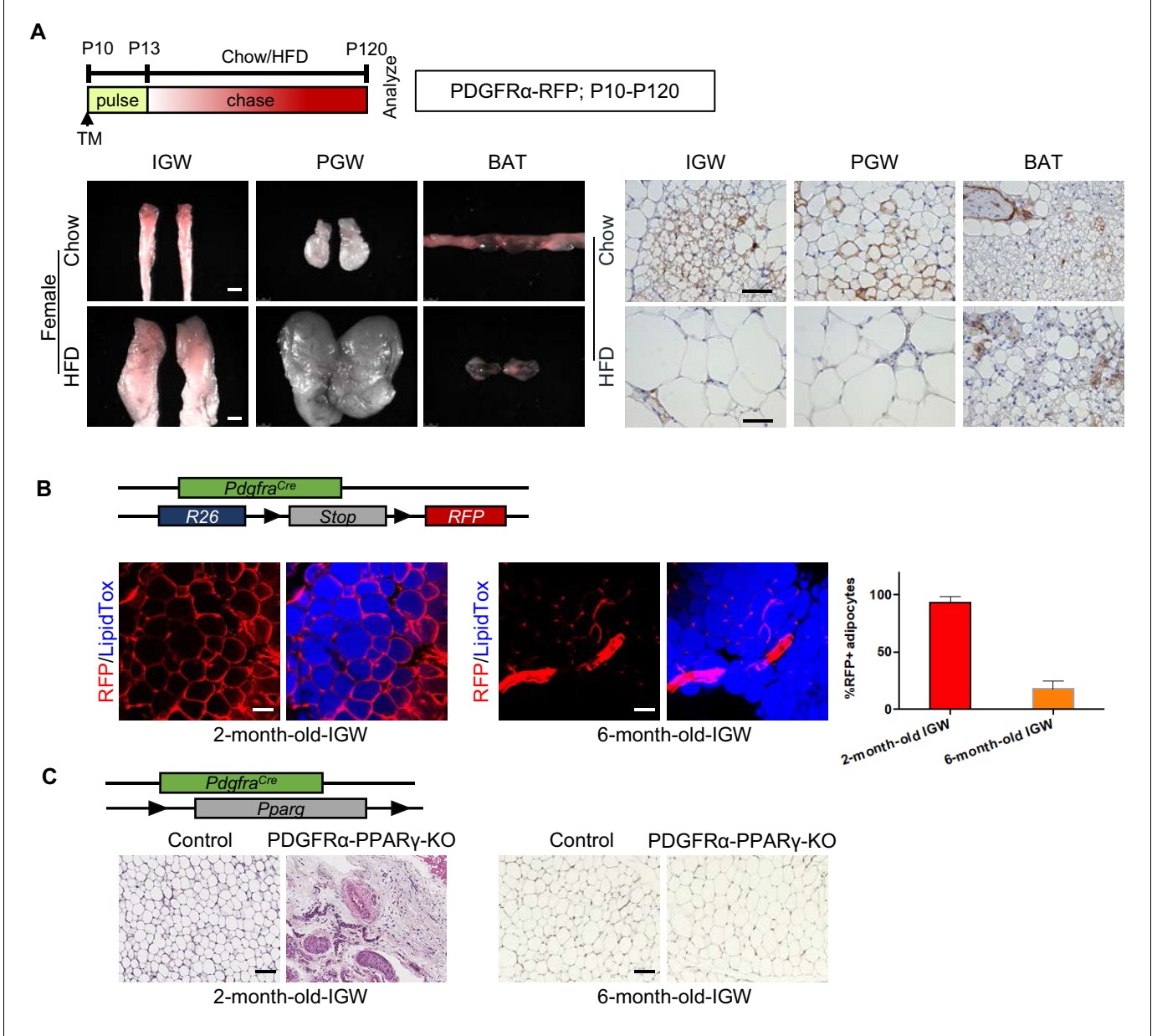

**Figure 3.** Developmental PDGFRα+ cells contribute to postnatal but not adult WAT development. (**A**) *Pdgfra^Cre-ERT2; Rosa26R^RFP* (PDGFRα-RFP) mice were administered TM (**A**) at postnatal day (**P**) 10 and fed chow or HFD until P120. IGW, PGW, and BATs were examined for direct RFP fluorescence and RFP IHC staining. Scale = 100 µm and 200 µm. (**B**) A 2- and 6-month-old PDGFRα-RFP mice were analyzed. IGWs were examined for direct RFP fluorescence and stained with LipidTox (blue). The quantifications for numbers of RFP+ adipocytes were calculated. Scale = 100 µm. (**C**) A 2- and 6-month-old control and *Pdgfra^Cre; Pparg^fl/fl* (PDGFRα-PPARγ-KO) mice were analyzed. IGWs were examined using hematoxylin and eosin (H&E) staining. Scale = 100 µm.

The online version of this article includes the following figure supplement(s) for figure 3:

**Figure supplement 1.** P10 PDGFRα+ cells contribute to postnatal but not adult WAT development.

(*Figure 4—figure supplement 1*). Histologically, we did not observe obvious phenotypic difference between control and mutant WAT or BAT morphology and architecture (*Figure 4F*). We also assessed fate-mapping analysis of control and SMA-PDGFRα-KO APCs to produce adipocytes. In line with our previous observations, control APCs generated RFP labeled adipocytes (*Figure 4G*). Similarly, SMA-PDGFRα-KO APCs also generated white adipocytes with the same efficiency (*Figure 4G*). Thus, our fate-mapping data support that new adipocytes generated from SMA+ cells

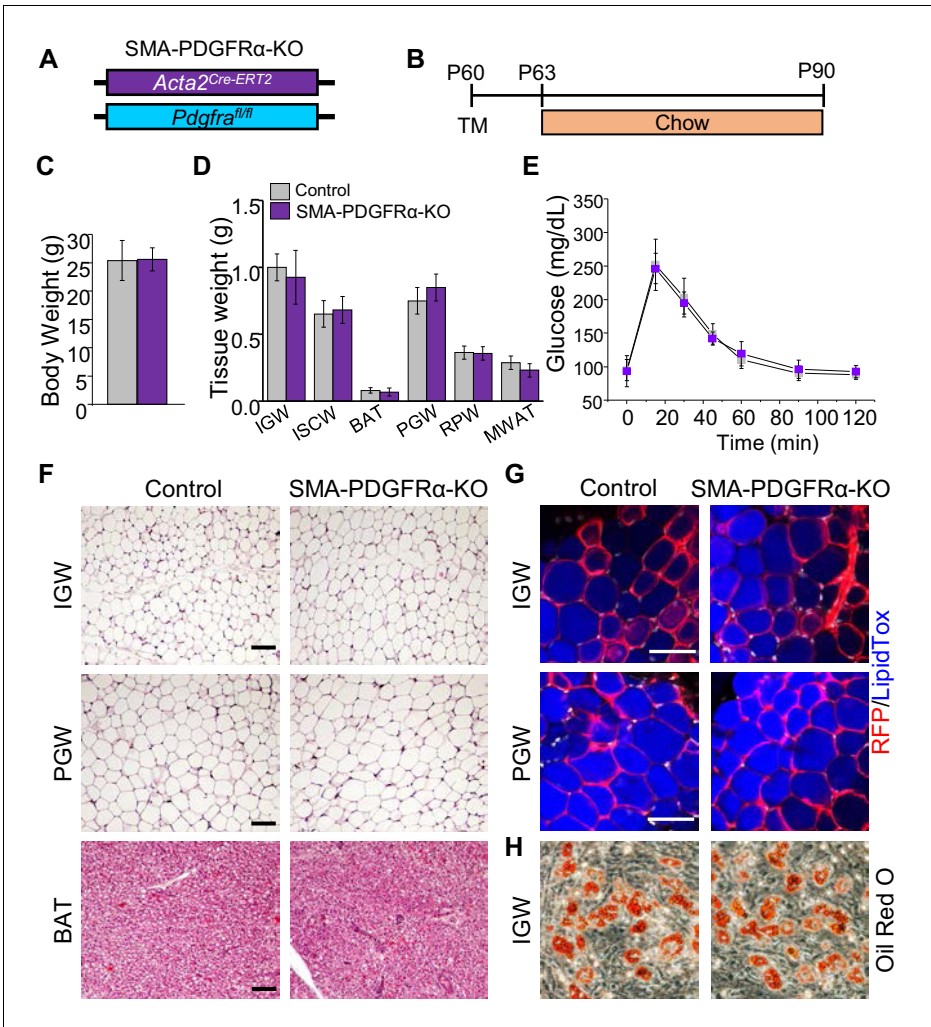

**Figure 4.** Deleting *Pdgfra* in adult SMA+ APCs is dispensable for adult white adipogenesis. (A–B) *Acta2^Cre-ERT2*; *Pdgfra^fl/fl* male control and mutant (SMA-PDGFRα-KO) mice were administered TM at P60 and analyzed at P90. (C) Body weight of control and SMA-PDGFRα-KO mice at P90. Data are expressed as mean ± SEM. (D) Adipose tissue weights. Data are expressed as mean ± SEM. (E) Blood glucose level during glucose tolerance test. Data are expressed as mean ± SEM. (F) Hematoxylin and eosin (H&E) staining of IGW, PGW, and BAT. Scale = 100 μm. (G) IGW and PGW were analyzed for direct RFP fluorescence and stained with LipidTox. Scale = 200 μm. (H) Oil Red O staining of SVF adipogenesis from IGW of control and SMA-PDGFRα-KO male mice at P90.

The online version of this article includes the following figure supplement(s) for figure 4:

**Figure supplement 1.** SMA-PDGFRα-KO mice do not display abnormal energy expenditure.

(RFP+) in both IGW and PGW are not affected by *Pdgfra* deletion. To exclude the possibility that SMA-PDGFRα-KO cells generated dysfunctional adipocytes, we examined metabolic performance using metabolic cage analysis of control and mutant mice at P90, after a 30 day chase. We observed that control and mutant mice showed similar energy expenditure, oxygen consumption, carbon dioxide production, and respiratory exchange ratio (*Figure 4—figure supplement 1*). Taken together, these data indicate that PDGFRα does not play a significant role in the ability of adult APCs to generate adipocytes and maintain adult WAT homeostasis.

To further evaluate if PDGFRα functioned in SMA+ APCs, we isolated SV cells from tamoxifen pulse control and SMA-PDGFRα-KO mice at P60 and subsequently culture them in adipogenic media. The adipogenic potential of SMA-PDGFRα-KO cells appeared similar to control SV cells as assessed by Oil Red O staining (*Figure 4H*).

Previous fate-mapping studies using *Acta2*<sup>Cre-ERT2</sup> revealed that SMA+ WAT resident perivascular cells also serve as beige progenitors: new beige adipocytes are formed in WAT rapidly when mice are exposed to cold, in part through de novo differentiation from SMA+ progenitors (*Berry et al., 2016b*). Therefore, we decided to examine if PDGFRα is required for beige adipogenesis using the SMA-PDGFRα-KO mouse model. We administered one dose of tamoxifen for 2 consecutive days to both control and mutant mice at P90 and then waited 2 weeks prior to cold exposure (6.5°C) (*Figure 5A*). Both control and mutant mice had a similar rectal temperature, a surrogate for beiging, at the end of cold exposure (*Figure 5B*). SMA-PDGFRα-KO mice had similar body weight, serum glucose level, and adiposity as controls after cold exposure (*Figure 5C–E*). Histologically, H&E staining and UCP1 IHC of IGW and PGW depots showed similar results in control and mutant mice (*Figure 5F,G*). There was also no significant BAT morphological difference between control and mutant mice (*Figure 5—figure supplement 1*). Consistent with H&E staining, qPCR analysis of thermogenic genes (*Ucp1*, *Prdm16*, and *Cidea*) from whole IGW depots suggested no significant

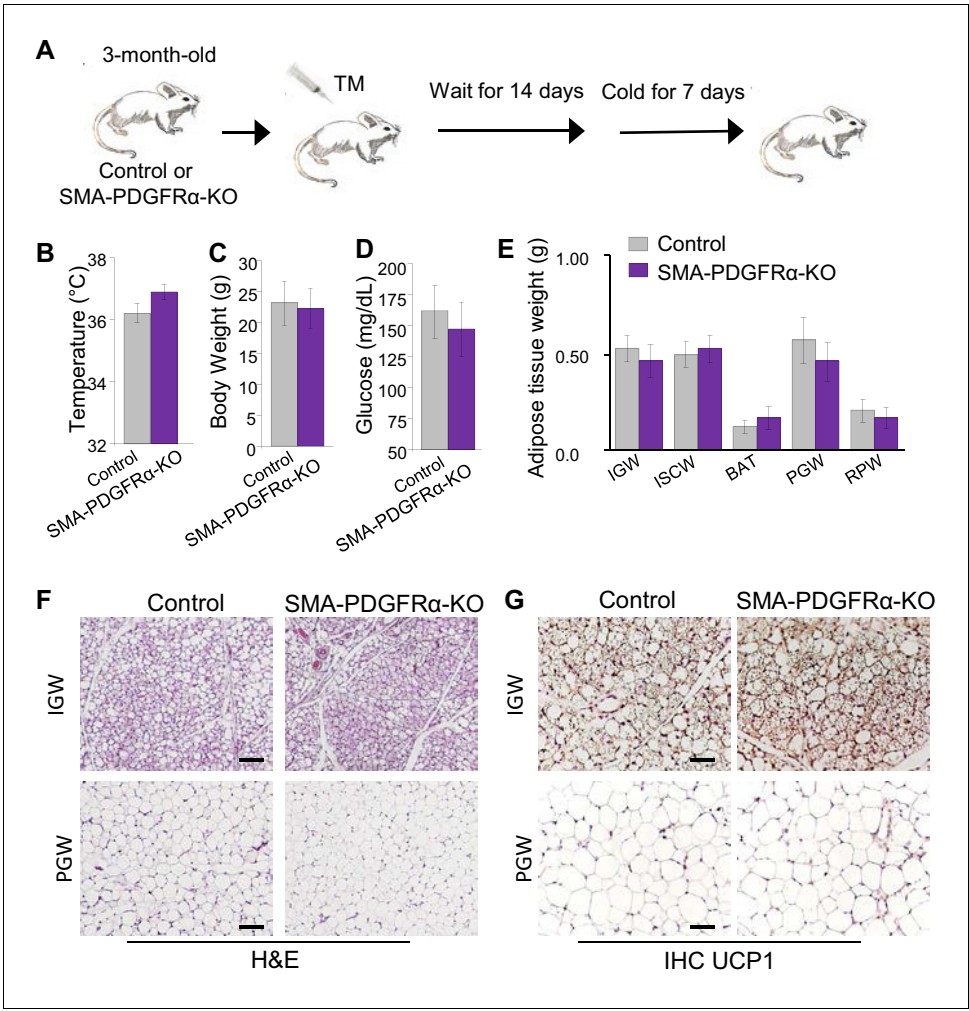

**Figure 5.** Deleting *Pdgfra* in adult SMA+ APCs is dispensable for cold-induced beige adipogenesis. (**A**) A 3-month-old *Acta2*<sup>Cre-ERT2</sup>; *Pdgfra*<sup>fl/fl</sup> male control and SMA-PDGFRα-KO mice were administered TM. After 14 days of TM washout, the mice were cold-exposed for 7 days. (**B**) Rectal temperature after cold exposure. Data are expressed as mean ± SEM. (**C**) Body weight after cold exposure. Data are expressed as mean ± SEM. (**D**) Blood glucose after cold exposure. Data are expressed as mean ± SEM. (**E**) Adipose tissue weight. Data are expressed as mean ± SEM. (**F**) Hematoxylin and eosin (H&E) staining of IGW and PGW. Scale = 100 μm. (**G**) UCP1 staining of IGW and PGW using immunohistochemistry (IHC). Scale = 100 μm.

The online version of this article includes the following figure supplement(s) for figure 5:

**Figure supplement 1.** Deleting *Pdgfra* in adult SMA+ APCs is dispensable for cold-induced beige adipogenesis.

difference between control and mutant beige adipocyte potential (*Figure 5—figure supplement 1*). These data suggest that PDGFRα may not have a functional role in adult SMA+ APCs in altering their ability to generate cold-inducible beige adipocytes.

## PDGFRα in developmental APCs is essential for adipose tissue development

Our data thus far suggests that PDGFRα does not have a functional role in adult adipogenic potential. Therefore, we decided to re-examine PDGFRα's role in WAT organogenesis and combined the AdipoTrak (AT) adipose lineage tracking and deletion system (*Pparg*$^{tTA}$; TRE-*Cre*; TRE-*H2B-GFP*) (*Tang et al., 2008*) with the *Pdgfra*$^{fl/fl}$ conditional mouse model (*Pparg*$^{tTA}$; TRE-*Cre*; TRE-*H2B-GFP*; *Pdgfra*$^{fl/fl}$ = AT-PDGFRα-KO) (*Figure 6A*). Of note, AdipoTrak labeled P10 and P30 cells express *Pdgfra* based on qPCR analysis (*Jiang et al., 2014*). Although control and AT-PDGFRα-KO mice displayed similar body weight at P60 (*Figure 6B*), AT-PDGFRα-KO mutant mice showed smaller adipose depots and reduced WAT weights (*Figure 6C,D*). By contrast, the weights of other tissues such as liver, kidney, spleen, pancreas, muscle, and heart showed no difference compared to controls (*Figure 6E*). Glucose tolerance test showed that mutant mice had impaired glucose tolerance, which may be due to the deficiency of functional adipocytes (*Figure 6F*). Histological staining revealed a paucity in adipocytes and only remnant adipocytes could be observed in mutant IGW and interscapular WAT (ISCW) (*Figure 6G*; *Figure 6—figure supplement 1*). Lipodystrophy is often accompanied by other metabolic disturbances such as liver steatosis; however, mutant mice did not appear to display fatty liver disease at this stage of life (*Figure 6—figure supplement 1*). We also evaluated the cell-autonomous adipogenic potential of SV cells. Specifically, SV cells were isolated from control and mutant mice and cultured in adipogenic media. Compared to control cells which are highly adipogenic, the AT-PDGFRα-KO mutant cells did not display adipogenic potential based on the appearance and Oil Red O staining (*Figure 6H*). These data strongly indicate that PDGFRα in developmental APCs is essential for adipose tissue development.

## PDGFRα regulates adipose tissue development

Our histological staining of AT-PDGFRα-KO WAT demonstrated the lack of adiposity with fibrotic tissue replacement; therefore, we tested if PDGFRα loss led to fibrosis. Trichrome collagen staining of IGW depots showed the presence of fibrotic tissue in mutant but not in control specimens (*Figure 7A*). We then assessed if APCs deficient in PDGFRα resulted in changes in APC locality and number. Whole-mount imaging of control and mutant WAT demonstrated the presence of GFP+ APCs in the correct anatomic anlage (*Figure 7B*). PDGFRα-deficient GFP+ cells also appeared to occupy the correct perivascular niche position (*Figure 7—figure supplement 1*). We then performed FACS analysis on GFP+ APC number and found AT-PDGFRα-KO mice had many more GFP+ progenitors than control WAT (control: 14.4% of SV cells; mutant: 59.9% of SV cells) (*Figure 7C–D*). Further analysis of these depots via FACS showed an increase in the endothelial marker *Cd31* (PECAM) (*Figure 7C*). Directed qPCR analyses of the FACS-isolated GFP+ cells from the AT-PDGFRα-KO mutant mice verified the reduction in *Pdgfra* mRNA expression. Mutant GFP+ cells had lower expression of adipogenic markers, including *Pparg, Fabp4, Plin1, and Lep* (*Figure 7E*). Consistent with the trichrome collagen staining, mutant GFP+ cells had higher expression of fibroblast markers, such as *Col1a1, Col3a1, Col6a1*, and *Ddr2*, compared to those from the control mice (*Figure 7F*). These data suggest that the loss of PDGFRα within the APC lineage promotes fibrotic gene expression rather than the adipogenic program. This could be a potential rationale for the fibrotic tissue incorporation in these WATs and could suggest a lineage fate switch (*Figure 7G*).

## Discussion

APCs are key components for WAT formation, maintenance, and expansion, and a variety of external stimuli can influence adipose homeostasis by controlling the regulatory mechanisms (*Berry et al., 2013*; *Berry et al., 2014*; *Hepler et al., 2017*; *Jiang et al., 2012*; *Sebo and Rodeheffer, 2019*; *Tang et al., 2008*). Distinct populations of APCs have been identified, but their relationship and the relevance to physiological and pathological adipose expansion remains unknown. We previously reported that there are two distinct adipose progenitor compartments, developmental and adult, which are utilized for adipose organogenesis and adipose homeostasis, respectively (*Jiang et al.,*

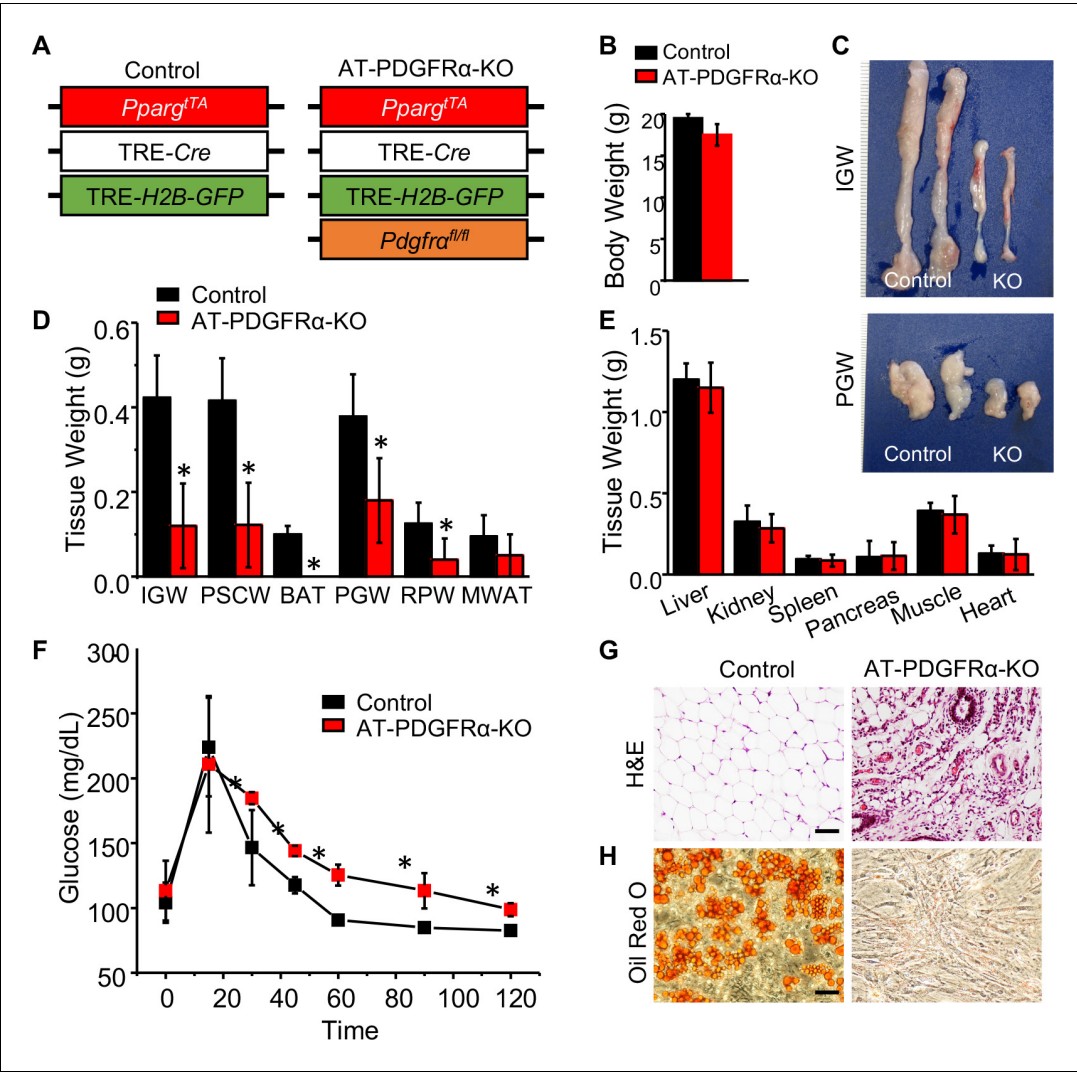

**Figure 6.** PDGFRα in developmental APCs is essential for adipose tissue development. (**A**) A 2-month-old *Pparg^{tTA}*; TRE-*Cre*; TRE-*H2B-GFP*; *Pdgfra^{fl/fl}* male control and AT-PDGFRα-KO mice were analyzed. (**B**) Body weight. Data are expressed as mean ± SEM. (**C**) IGW and PGW tissue. (**D**) Adipose tissue weight. *p<0.05 AT-PDGFRα compared to AT-Con mice. Data are expressed as mean ± SEM. (**E**) Other tissue weight. Data are expressed as mean ± SEM. (**F**) Blood glucose level during glucose tolerance test. *p<0.05 AT-PDGFRα-KO compared to control mice. Data are expressed as mean ± SEM. (**G**) Hematoxylin and eosin (H&E) staining of IGW. Scale = 100 μm. (**H**) Oil Red O staining of SVF isolated from IGW of control and AT-PDGFRα-KO mice.

The online version of this article includes the following figure supplement(s) for figure 6:

**Figure supplement 1.** PDGFRα in developmental APCs is essential for adipose tissue development.

*2014*). We have demonstrated that these two different progenitor pools have different microanatomical, functional, morphological, genetic, and molecular profiles. Notably, adult progenitors fate map from a SMA+ mural cell lineage while developmental progenitors do not (*Jiang et al., 2014*). However, the identity of developmental APCs and the regulatory mechanisms governing WAT development and homeostasis were unclear. In the current study, we attempted to disentangle these two APC populations by using a PDGFRα tamoxifen-inducible lineage-tracing system. We found that PDGFRα+ cells generate adipocytes during development but not during adult WAT homeostasis. Further, we show that the role of PDGFRα is differentially required. For example, during development, PDGFRα signaling is important for APCs to generate mature adipocytes. On the other hand, during WAT homeostasis, PDGFRα signaling in SMA+ adult APCs is dispensable for both white and cold-induced beige adipocyte formation. Our results implicate PDGFRα as a

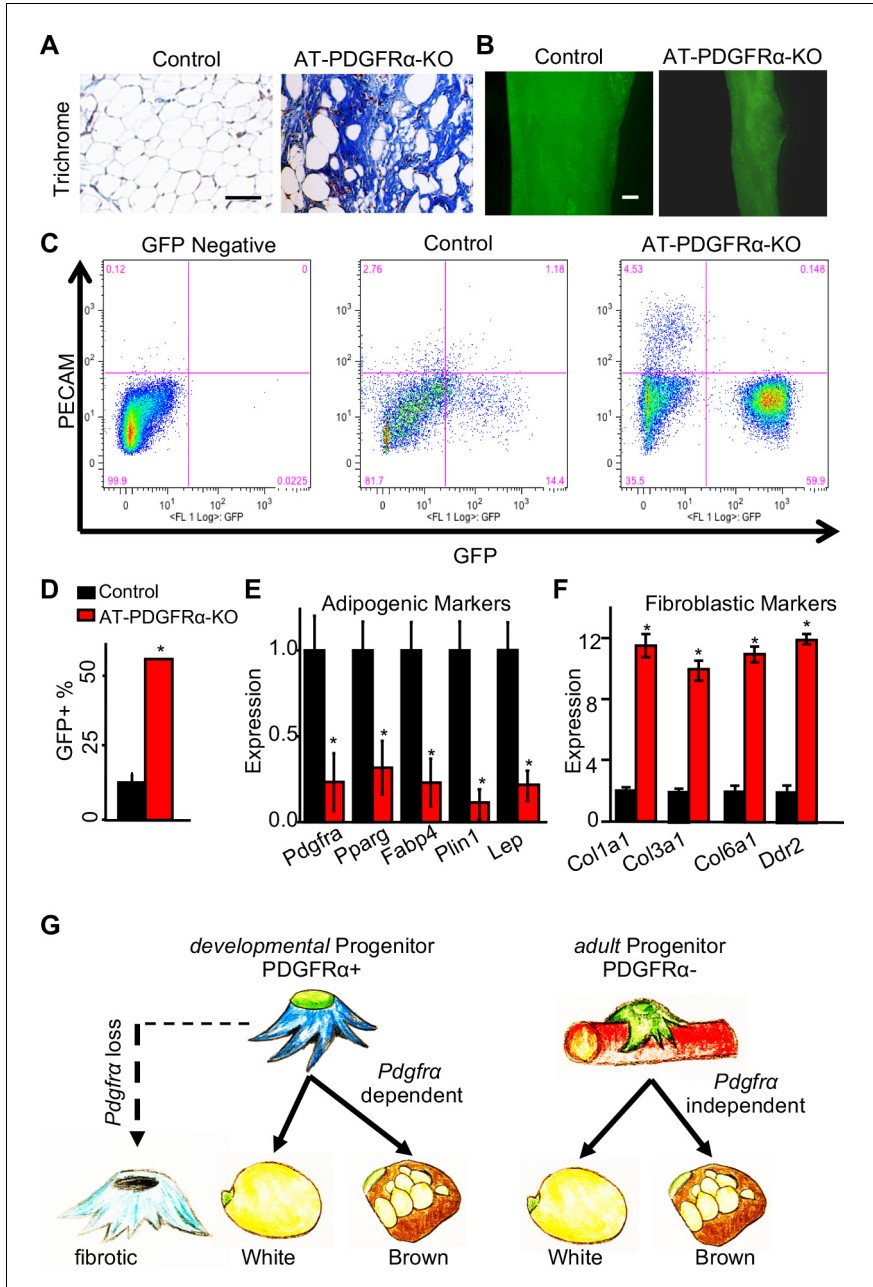

**Figure 7.** PDGFRα regulates adipose tissue development through lineage control. (**A**) A 2-month-old *Pparg^{tTA}*; TRE-*Cre*; TRE-*H2B-GFP*; *Pdgfra^{fl/fl}* male control and AT-PDGFRα-KO mice were analyzed. Trichrome staining of IGW. Scale = 200 μm. (**B**) Direct GFP fluorescence of IGW. Scale = 100 μm. (**C**) Flow cytometry profiles of SVF isolated from IGW. (**D**) Quantification of GFP+ adipose progenitor cell number. Data are expressed as mean ± SEM. (**E**) Adipogenic marker gene expression levels in GFP+ cells of SVF isolated from IGW. Data are expressed as mean ± SEM. (**F**) Fibrotic marker gene expression levels in GFP+ cells of SVF isolated from IGW. Data are expressed as mean ± SEM. *p<0.05 AT-PDGFRα-KO compared to control mice in D-F. (**G**) Working model for PDGFRα in developmental and adult progenitors. Developmental progenitors are marked by PDGFRa+ and adipogenesis is dependent on PDGFRα. In the absence of PDGFRα, developmental progenitors switch the lineage to fibrotic. Adult progenitors for WAT homeostasis are not marked by PDGFRα+ and adipogenesis during WAT homeostasis is largely PDGFRα independent.

The online version of this article includes the following figure supplement(s) for figure 7:

**Figure supplement 1.** *Pdgfra* deletion in developmental APCs.

regulator of developmental APC lineage specification and adipogenesis in turn promoting WAT organ development.

Our study provides fate-mapping and genetic evidence that PDGFRα is differentially required for adipogenesis at different times of lifespan, revealing the distinct regulatory mechanisms governing adipose tissue development versus adult adipose tissue homeostasis in vivo. Other studies also suggest the existence of distinct regulatory mechanisms for WAT development and maintenance (*Shao et al., 2017*; *Wang et al., 2015*). It has been reported that CEBPA (CCAAT Enhancer Binding Protein Alpha), a critical transcription factor expressed during adipogenesis, is not required for WAT development and maintenance during the fetal and early postnatal stage, but it is essential for the obesogenic expansion of WAT induced by HFD (*Wang et al., 2015*). ZFP423 (zinc finger protein 423), expressed in committed preadipocytes (*Gupta et al., 2012*), is shown to regulate adipocyte differentiation during fetal adipose development; however, in adult mice, it controls a white-to-beige phenotypic switch (*Shao et al., 2017*). Also, *Adipoq* driven Cre is more actively expressed at an earlier stage of the adipocyte life cycle during fetal WAT development compared to adult mice (*Shao et al., 2017*). AKT2 (Serine/Threonine Kinase 2) is dispensable for adipose tissue development but required for CD24+ adipose progenitor cell proliferation in postal animals (*Jeffery et al., 2015*). These studies consistently indicate that developmental and adult APCs utilize distinct regulatory mechanisms to respond to developmental and nutritional cues. The findings may also hint that adipocytes generated during developmental and adult stages have distinct functions.

Our data suggest that PDGFRα signaling, specifically in SMA+ APCs, does not play a significant role in SMA+ progenitors' differentiation into adipocytes under physiological condition. It is unclear at this point whether PDGFRα signaling in adult APCs has other important functions under pathological conditions. For example, PDGFRα signaling might be essential for the obesogenic expansion and fibrotic response of WAT induced by HFD. This hypothesis, while speculative, is consistent with several studies reported that overexpressing PDGFRα inhibits adipogenesis and promotes fibrosis (*Hepler et al., 2017*; *Iwayama et al., 2015*; *Lee et al., 2012*; *Marcelin et al., 2017*; *Sun et al., 2017*). A more specific study showed that a subset of PDGFRα+ cells with high CD9 expression, induced by obesity, originates pro-fibrotic cells, while those with low CD9 expression are committed to adipogenesis (*Iwayama et al., 2015*; *Sun et al., 2017*). HFD feeding triggers the recruitment of PDGFRα+ cells and obesity induces CD9 expression in PDGFRα+ cells, which become fibrotic cells (*Marcelin et al., 2017*). Thus, we cannot completely rule out the roles of PDGFRα signaling in adult APCs unless a proadipogenic and a fibrogenic challenge is done in the mice and fibrosis is assessed. It will be interesting to challenge SMA-PDGFRα-KO mice with HFD feeding and then test if PDGFRα in adult APCs plays a role in obesogenic WAT expansion.

Several groups have now reported that adipose stromal cells express PDGFRα (*Hepler et al., 2017*; *Iwayama et al., 2015*; *Lee et al., 2012*; *Marcelin et al., 2017*; *Sun et al., 2017*). In addition, fate-mapping studies using inducible *Pdgfra-Cre* have been performed in multiple labs. However, the results are strikingly different. For example, a more recent study using *Pdgfra-MerCreMer* lineage traced animals found that PDGFRα+ fibroblasts gave rise to brown, beige, and white adipocytes during adult homeostasis (*Cattaneo et al., 2020*). Another study suggests that PDGFRα+ cells are bi-potential to produce both beige and HFD-induced white adipocytes (*Lee et al., 2012*). Moreover, another study suggests the balance between PDGFRα/PDGFRβ signaling determines progenitor commitment to beige or white adipogenesis (*Gao et al., 2018*). Multiple factors may account for this discrepancy. One possible explanation lies in the use of different inducible *Cre* models of *Pdgfra*. The *Pdgfra*-MerCreMer was produced by knocking in the inducible *Cre* cassette into the endogenous *Pdgfra* locus, which represents native *Pdgfra* expression; yet, it may disrupt *Pdgfra* transcription (*Ding et al., 2013*). By contrast, our *Pdgfra*<sup>Cre-ERT2</sup> line was generated using BAC transgenic mice (*Rivers et al., 2008*), which did not affect endogenous expression but may not fully recapitulate the endogenous *Pdgfra* expression. Another potential factor in the difference is the *Cre-loxP* recombination efficiency and cell-type specificity. In a previous report (*Jiang et al., 2017a*), we have characterized the *Pdgfra*<sup>Cre-ERT2</sup> model and showed that PDGFRα-RFP+ cells were 100% positive for PDGFRα antibody staining, indicating this system can specifically label the cell types of interest, which is PDGFRα+. In the current study, our FACS analysis showed less PDGFRα-RFP labeling at P60 compared to P10 WAT labeling. From our studies, it is unclear why there is less labeling; however, one notion we support is an overall total reduction in PDGFRα expressing cells within WATs. An additional exploration into this hypothesis would be critical for evaluating PDGFRα

expression and function in future studies. We also noted, at P10, RFP- cells have about a third of the mRNA of *Pdgfra* than RFP+ cells, indicating not all of the PDGFRα+ cells underwent DNA recombination under our experimental conditions and labeling efficiency. This low labeling is consistent with a previous study by Jeffery et al, who reported that two different inducible lines of *Pdgfra*[Cre-ERT2] marked adipose lineage with variable efficiency (*Jeffery et al., 2014*). In alignment, Rodeheffer and colleagues demonstrated the lack of significant fate mapping of PDGFRα+ cells to de novo adult WAT adipogenesis (*Jeffery et al., 2014*). Nevertheless, our fate-mapping data using this *Pdgfra*[Cre-ERT2] transgenic line suggest a previously unanticipated differential labeling and contribution of PDGFRα in WAT development and homeostasis. A likely reason we postulate for the differential contribution of PDGFRα is possibly due to the dynamic expression of PDGFRα between developmental and adult APCs.

A limitation of our studies is that our developmental fate-mapping data did not allow us to discriminate whether functionally distinct PDGFRα+ cells exist to give rise to white and brown adipocytes, or if there is a common progenitor for all adipocytes in different depots. Recent studies using single-cell RNA-sequencing reveal that distinct subpopulations of APCs in the stromal vascular fraction of WAT are present in both mouse and human adipose tissues (*Burl et al., 2018*; *Gu et al., 2019*; *Hepler et al., 2018*; *Merrick et al., 2019*; *Schwalie et al., 2018*; *Zhou et al., 2019*). It will be of future interest to perform single-cell RNA-sequencing or clonal lineage tracing to examine the heterogeneity PDGFRα+ cells and their relationship to other APCs within a single depot.

In summary, our study suggests that PDGFRα signaling plays a key role in adipose tissue development by determining adipose progenitor cell fate and in the regulation of progenitor cell dynamics under the HFD challenge. This study expands the current knowledge regarding independent adipose progenitor compartments for WAT formation and maintenance. These data highlight the key roles of the distinct APC and their different regulatory mechanism governing WAT organogenesis and homeostasis. Our results may help to discover the new therapeutic targets for treatment and prevention of both childhood and adult obesity, and the subsequent metabolic dysregulation.

# Materials and methods

## Animals

All animals were maintained under the guidelines of the University of Illinois at Chicago (UIC) Animal Care and Use Committee. Mice were housed in a 14:10 light:dark cycle, and experimental diets and water were provided ad libitum. AdipoTrak mice are defined as *Pparg*[tTA]; TRE-*Cre* (JAX Stock: 006234); TRE-*H2B-GFP* as previously reported (*Tang et al., 2008*). *Rosa26R*[RFP] (JAX Stock No: 007908), *Pdgfra*[Cre] (JAX Stock No: 013148), *Pparg*[fl/fl] (JAX Stock No: 004584), and *Pdgfra*[fl/fl] (JAX Stock No: 006492) mice were obtained from the Jackson Laboratory. *Acta2*[Cre-ERT2] mice were generously provided by Dr. Pierre Chambon. Drs. Sean Morrison and Bill Richardson generously provided the *Pdgfra*[Cre-ERT2]. Cre recombination was induced by administering one dose of tamoxifen (Cayman, Ann Arbor, MI) dissolved in sunflower oil (Sigma-Aldrich, St. Louis, MO) for 1 or 2 consecutive days (50 mg/Kg intraperitoneal injection). In these experiments, tamoxifen was given to all animal groups including control mice which carried the floxed alleles but lacked the *Cre* transgene. For cold exposure, mice were placed in a 6.5°C cold metabolic chamber for 7 days. The mice were fed either normal chow (4% Kcal fat, Harlan-Teklad, Madison, WI) or high-fat diet (HFD; 60% Kcal fat; D12492, Research Diets, New Brunswick, NJ). Rosi intake was estimated to be 15 mg/kg body mass/day.

## Stromal vascular fraction isolation and cell culture

Stromal vascular (SV) cells were isolated from subcutaneous WAT, including inguinal and interscapular depots. After 1 hr of slow shaking of the tissue in isolation buffer (100 mM HEPES pH 7.4, 120 mM NaCl, 50 mM KCl, 5 mM Glucose, 1 mM $CaCl_2$, and 1.5% BSA) containing 1 mg/mL Collagenase, Type 1 (Worthington Biochemical Corporation, Lakewood, NJ) at 37°C, the suspension was centrifuged at 1000 × g for 10 min. The floating adipocyte layer and the solution were removed, and the SV pellet was resuspended in DMEM/F12 media (Sigma-Aldrich, St. Louis, MO) supplemented with 10% FBS (Sigma-Aldrich, St. Louis, MO) and 1% Penicillin/Streptomycin (Gibco, Waltham, MA).

The isolated mouse SV cells were cultured in DMEM/F12 media (Sigma-Aldrich, St. Louis, MO) supplemented with 10% FBS (Sigma-Aldrich, St. Louis, MO) and 1% Penicillin/Streptomycin (Gibco, Waltham, MA). For Cre induction in the cell culture system, we used 2 uM 4-hydroxytamoxifen (4-OH-tamoxifen, Sigma, Sigma-Aldrich, St. Louis, MO). White adipogenesis was induced by treating confluent cells with DMEM/F12 containing 10% FBS, 1 µg/mL insulin (Sigma-Aldrich, St. Louis, MO), 1 µM dexamethasone (Cayman, Ann Arbor, MI), and 0.5 mM isobutylmethylxanthine (Sigma-Aldrich, St. Louis, MO) for the first 3 days and with DMEM/F12 containing 10% FBS and 1 µg/mL insulin for another 3 days.

## Flow cytometry

SV cells were isolated, washed, centrifuged at $1000 \times g$ for 10 min, and sorted with a MoFlo Astrios Cell Sorter (Beckman Coulter, Brea, CA) operated by the UIC Flow Cytometry Core. For RFP+ sorting, live SV cells from P10 and P60 tamoxifen-injected $Pdgfra^{Cre-ERT2}$; $Rosa26R^{RFP}$ mice were stained with DAPI to exclude dead cells and sorted based on native fluorescence. The SV cells from RFP- mice were used to determine background fluorescence levels. For GFP+ and RFP+ flow analysis, SV cells were isolated from $Pparg^{tTA}$; TRE-H2B-GFP; $Pdgfra^{Cre-ERT2}$; $Rosa26R^{RFP}$ double reporter mice (IGW, PGW, and BAT were pooled from n = 10 for P10 and n = 8 for P60 mice). For GFP+ (native fluorescence)/CD31+ flow analysis, SV cells were isolated from P30 Control and AT-PDGFRα-KO mice. SV cells were stained with rat anti-CD31 (CD31;1:200 BD Bioscience: item no: 550274) on ice for 30 min. Cells were then washed twice with the staining buffer and incubated with cy5 donkey anti-rat (1:500, Jackson ImmunoResearch, item no: 711-605-152) secondary antibody for CD31. Cells were incubated for 30 min on ice before flow cytometric analysis. For gating strategies of both GFP sorting and flow analysis, live cells were selected by size on the basis of FSC and SSC. Single cells were then gated on both SSC and FSC Width singlet's. SVF cells isolated from GFP-negative mice, along with primary-minus-one controls, were used as a negative control to determine background fluorescence levels.

## Real-time quantitative PCR

Total RNA was extracted using TRIzol (Invitrogen, Carlsbad, CA) from either mouse tissues or cells. cDNA was synthesized using High Capacity RNA to cDNA Kit (Life Technologies, Carlsbad, CA), and gene expression was analyzed using Power SYBR Green PCR Master Mix with ViiA7 Real-time PCR System (Applied Biosystems, Foster City, CA). Quantitative PCR values were normalized to 18 s rRNA expression. Primer sequences are provided in *Supplementary file 1*.

## Histological staining

Hematoxylin and eosin (H&E) or trichrome staining was carried out on paraffin sections using the following procedure. Adipose tissues were fixed in 10% formalin overnight, processed in STP120 tissue processing unit (Thermo-Fisher Scientific, Waltham, MA) in a series of ethanol dehydrated steps (50%, 70%, 80%, 95%, 95%, 100%, and 100% at 45 min/step) and xylene substitute rinse steps (three times, 45 min/step), and then submerged in paraffin (two times, 1 hr/step). Processed tissues were embedded in paraffin blocks using a HistoStar tissue embedding station (Thermo-Fisher Scientific, Waltham, MA), and the embedded tissues were sectioned with an HM325 microtome (Thermo-Fisher Scientific, Waltham, MA) at 8 to 12 µm thickness. Slides were baked for 1 hr at 55°C and stained with H&E. For immunohistochemistry (IHC), sections were deparaffinized, boiled in antigen-retrieval solution, treated with UCP1 antibody (1:500, ab23841, Abcam, Cambridge, United Kingdom), and stained with Vectastain ABC KIT (PK-6100, Vector Laboratories, Burlingame, CA) and DAB KIT (SK-4100, Vector Laboratories, Burlingame, CA). RFP (tdTomato) reporter expression in paraffin sections was visualized by immunostaining with a mouse monoclonal antibody against DsRed (Takara, 632392) at 1:500. Adipocytes were identified by immunostaining with anti-Perilipin-1 (Abcam, ab61682) used at 1:1000. To stain lipid, chopped adipose tissues were incubated in LipidTox (Invitrogen, Carlsbad, CA) at 1:200 in PBS for overnight at 4°C before washing in 1× PBS and mounting for imaging. Whole-mount images were taken on a Leica M205 FA microscope, and immunostaining images were collected on a Leica DMi8 inverted microscope. For quantification of images, two independent observers assessed three random fields in 10 random sections from at least three mice per cohort.

## Oil Red O staining

In vitro differentiated cells were fixed in 4% paraformaldehyde for 45 min at room temperature. After washing with 1× PBS twice, the cells were stained with Oil Red O working solution (0.5% iso-propanol, Sigma-Aldrich, St. Louis, MO) at room temperature for 30 min. The Oil Red O solution was removed, and the cells were washed with 1× PBS before imaging.

## Metabolic phenotyping experiments

Temperature was monitored daily using a rectal probe (Physitemp). The probe was lubricated with glycerol and was inserted 1.27 cm (0.5 in), and the temperature was measured when stabilized. Body composition was measured using a Bruker Minispec 10 whole body composition analyzer (Bruker, Billerica, MA) at the UIC Biologic Resources Laboratory. For glucose monitoring, tail blood was drawn in the morning and blood glucose levels were measured with a Contour glucometer (Bayer, Leverkusen, Germany). For glucose tolerance tests, 1.25 mg glucose/g body weight of the mouse was injected intraperitoneally after a 5 hr fasting, and blood glucose levels were measured at the indicated intervals.

## Metabolic cage studies

Control and SMA-PDGFRα-KO mice were housed individually and acclimatized to the metabolic chambers at the UIC Biologic Resources Laboratory for 2 days before data collection was initiated. For the subsequent 3 days, food intake, $VO_2$, $VCO_2$, and physical activity were monitored over a 12 hr light/dark cycle with food provided ad libitum.

## Quantification and statistical analysis

All labeling quantifications were performed in at least four animals with a minimal of 3 distinct sections being imaged and counted per animal. A two-tailed Student's t-test or one-way ANOVA followed by post-hoc comparisons using the Bonferroni post-hoc test was conducted. A $p < 0.05$ was considered statistically significant. Data were presented as mean ± standard error of the mean (SEM) and plotted in GraphPad Prism 8.0. All experiments were performed on 2–3 independent cohorts with a minimum of 4 mice/group.

## Acknowledgements

We thank Dr. Jonathan M Graff for supporting this work and giving fruitful advice. We thank Dr. Pierre Chambon for the $Acta2^{Cre-ERT2}$ mouse strain. We thank Drs. Cynthia Rose Adams and Jeanette Purcell for assistance with mouse husbandry, Dr. Stefan J Green and the Research Resources Center for quantitative real-time PCR analysis, Dr. Brian Layden and Metabolic Phenotyping Core for analytical and phenotypical mouse measurements, Dr. Balaji Ganesh and Flow Cytometry Core facility for FACS and members of the Jiang laboratory for helpful comments on the manuscript. This work is supported to YJ by the National Institute of Diabetes and Digestive and Kidney Disease grant K01 DK111771 and Pilot and Feasibility Diabetes Research and Training Center (DRTC) Award (P30DK020595). DCB is supported by the National Institute of Diabetes and Digestive and Kidney Disease grant K01 DK109027.

## Additional information

### Funding

| Funder | Grant reference number | Author |
|---|---|---|
| National Institute of Diabetes and Digestive and Kidney Diseases | K01 DK111771 | Yuwei Jiang |
| Chicago Diabetes Research and Training Center | P30DK020595 | Yuwei Jiang |
| National Institute of Diabetes and Digestive and Kidney Diseases | K01 DK109027 | Daniel C Berry |

The funders had no role in study design, data collection and interpretation, or the decision to submit the work for publication.

## Author contributions
Sunhye Shin, Data curation, Formal analysis, Investigation, Visualization, Writing - original draft, Project administration, Writing - review and editing; Yiyu Pang, Pingwen Xu, Investigation, Writing - review and editing; Jooman Park, Data curation, Formal analysis, Investigation, Visualization; Lifeng Liu, Brandon E Lukas, Investigation, Visualization; Seung Hyeon Kim, Ki-Wook Kim, Investigation; Daniel C Berry, Conceptualization, Writing - review and editing; Yuwei Jiang, Conceptualization, Data curation, Formal analysis, Funding acquisition, Validation, Investigation, Writing - original draft, Project administration, Writing - review and editing

## Author ORCIDs
Sunhye Shin https://orcid.org/0000-0002-5916-5390
Jooman Park https://orcid.org/0000-0001-5368-5769
Daniel C Berry http://orcid.org/0000-0002-5200-1182
Yuwei Jiang https://orcid.org/0000-0002-5082-8012

## Ethics
Animal experimentation: This study was performed in strict accordance with the recommendations in the Guide for the Care and Use of Laboratory Animals of the National Institutes of Health. All of the animals were handled according to approved institutional animal care and use committee (IACUC) protocols (#18-184) of the University of Illinois at Chicago. The protocol was reviewed in accordance with the Animal Care Policies and Procedures of the University of Illinois at Chicago and renewed on 10/16/2019. All experimental animals will be euthanized by carbon dioxide gas inhalation in accordance with the guidelines of the American Veterinary Medical Association and the policies of the UIC IACUC.

## Decision letter and Author response
Decision letter https://doi.org/10.7554/eLife.56189.sa1
Author response https://doi.org/10.7554/eLife.56189.sa2

# Additional files

## Supplementary files
• Supplementary file 1. Primer sequences used.
• Transparent reporting form

## Data availability
All data generated or analyzed during this study are included in the manuscript and supporting files.

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
