## [Decision Letter]

**Acceptance summary:**

In this manuscript, the authors present a very interesting and high-quality body of work in a relevant topic for adipose and developmental biology fields. The mouse models used, and the experiments performed are elegant and imaginative. The author's findings indicate that PDGFRα signaling is required for WAT development but dispensable for adult WAT homeostasis.

**Decision letter after peer review:**

Thank you for submitting your article "Dynamic control of adipose tissue development and tissue homeostasis by platelet-derived growth factor receptor alpha" for consideration by *eLife*. Your article has been reviewed by two peer reviewers, and the evaluation has been overseen by a Reviewing Editor and Kathryn Cheah as the Senior Editor. The following individuals involved in review of your submission have agreed to reveal their identity: Stephen Farmer (Reviewer #1); Juan Sanchez-Gurmaches (Reviewer #2).

This manuscript presents a very interesting and high-quality body of work in a relevant topic for developmental biology and the adipose field. The authors use mouse models to perform elegant and imaginative experiments that reveal divergent progenitor cells for adipose tissue in development and adult. The reviewers have discussed the reviews with one another and the Reviewing Editor has drafted this decision to help you prepare a revised submission.

Summary:

This manuscript presents a very interesting and high-quality body of work in a relevant topic for the field that should be published. The mouse models used, and the experiments performed are elegant and imaginative.

1) The authors say that adult APC don't express PDGFRα. This is mainly based on the lack of activation of an inducible cre driver by tamoxifen in >P60 animals (Figure 2—figure supplement 1C and related Figure F and B (there are almost no cells labelled)). It was reported previously by Jeffery et al., 2014 that two PDGFRα-Cre^ERT2^ models (distinct from the one used here, as far as I can tell based on the Materials and methods section) didn't label APC. This paper is not mentioned in the reference list. Regarding the references that the authors use for validation of using PDGFRα cre drivers (subsection “Developmental adipocytes derive from a PDGFRα+ cell source”, second paragraph): Berry et al., 2013 don't use inducible drivers, and the Lee at al., 2012a paper uses a distinct inducible cre driver. Besides this, the authors don't directly compare the levels of expression (by RNA or protein (flow or other)) of PDGFRα between young and adult APC. Also, when the authors delete PDGFRα in adult APC with SMA-CreERT2 there is a quiet important decrease in PDGFRα mRNA in an unidentified tissue (it is not mentioned in the figure legend) suggesting that adult APC do express PDGFRα that can be deleted. Also, in Figure 2D, even the RFP- cells have as much as a third of the mRNA of PDGFRα than RFP+ cells in young animals. Also, a paper cited by the authors uses a MerCreMer approach for inducible labelling of PDGFRα expressing cells (Cattaneo et al., 2020) and the results contradict the ones in this manuscript. May all be a problem with the PDGFRα-CRE^ERT2^ model? May it not get enough Cre activity when the mice are old by the normal tamoxifen treatments? The authors should comment more in their Discussion about these caveats and if possible, directly compare young and adult APC in these regards.

2) The authors suggest that PDGFRα signaling have no function in adult APC. The results fall short for this affirmation as the authors already point out in the Discussion because we cannot be sure of this unless a proadipogenic and a fibrogenic challenge is done in the mice and fibrosis is actually assessed. I consider that shifting their conclusion and figure legend title to directly reflect their very interesting results would be a good idea.

Revisions expected in follow-up work:

3) Do PPARγtTA; TRE-Cre; TRE-H2B-GFP; PDGFRα^fl/fl^ mice recover normal fat mass by P120 as the model put forward suggests?

4) The experiment in Figure 3C is definitive. Do the authors want to mention why there is not a more thorough analysis of these mice?

5) Figure 5 needs additional analysis of the browning status of the available tissues to fulfil the conclusion. As of now we don't know if both genotypes have same levels of UCP1 in white and brown fats.

---

## [Author Response]

Revisions for this paper:1) The authors say that adult APC don't express PDGFRα. This is mainly based on the lack of activation of an inducible cre driver by tamoxifen in >P60 animals (Figure 2—figure supplement 1C and related Figure 2F and B (there are almost no cells labelled)). It was reported previously by Jeffrey et al., 2014 that two PDGFRα-Cre^ERT2^ models (distinct from the one used here, as far as I can tell based on the Materials and methods section) didn't label APC. This paper is not mentioned in the reference list.

This was an unfortunate oversight on our part, and we thank the reviewer for catching it. The citation is now appropriately included the paper (Jeffery et al., 2014) in the Discussion section (fourth paragraph).

Regarding the references that the authors use for validation of using PDGFRα cre drivers (subsection “Developmental adipocytes derive from a PDGFRα+ cell source”, second paragraph): Berry et al., 2013 don't use inducible drivers, and the Lee et al., 2012a paper uses a distinct inducible cre driver.

We apologize about the confusion with the references. In this revised manuscript, we have now corrected the reference (subsection “Developmental adipocytes derive from a PDGFRα+ cell source”, second paragraph).

Besides this, the authors don't directly compare the levels of expression (by RNA or protein (flow or other)) of PDGFRα between young and adult APC.

We thank the reviewers for pointing this out and we have now expanded the Discussion in our revised manuscript (fourth paragraph). The reason we did not do the experiments is because the levels of Pdgfrα expression between developmental and adult APCs have been reported in our previous publication Jiang et al., 2014, Figure 4K and Supplementary Figure 4B), using Pparγ-tTA; TRE-H2B-GFP AdipoTrak system. We have reported that FACS-isolated developmental APCs had lower levels of Pdgfrα expression at mRNA level (Figure 4K) compared to their adult counterpart, and flow analysis showed that the number of Pdgfrα positive cells were similar in developmental and adult PPARγ+ expressing APCs (Supplementary Figure 4B).

Also, when the authors delete PDGFRα in adult APC with SMA-Cre^ERT2^ there is a quiet important decrease in PDGFRα mRNA in an unidentified tissue (it is not mentioned in the figure legend) suggesting that adult APC do express PDGFRα that can be deleted.

We believe that there is some PDGFRα expression in SMA+ cells, which has been reported in our previous published studies Jiang et al., 2014, Figure 3G and H). Flow analysis indicated that about half of the RFP labeled SMA+ cells expressed PDGFRα. In addition, quantitative PCR analysis showed that Pdgfrα expression was ~10-fold enriched in RFP labeled SMA+ cells. Therefore, we would expect that our SMA-Cre^ERT2^; Pdgfrα^fl/fl^ deletion model reduces Pdgfrα mRNA significantly. The tissue used for Figure 4—figure supplement 1C analyses have been added to the figure legend (Figure 4—figure supplement 1C) and text (subsection “PDGFRα in adult SMA+ APCs is not required for adult white or beige adipogenesis under physiological conditions”). Together, these data suggest that only a subset of SMA+ cells resemble APCs and the ones co-express SMA and PDGFRα may not behave as APCs. We have included this previous data in the text.

Also, in Figure 2D, even the RFP- cells have as much as a third of the mRNA of PDGFRα than RFP+ cells in young animals.

We thank the reviewers for their careful attention to this issue. Indeed, we notice the RFP- cells still have Pdgfrα expression. We think this is highly likely due to the incomplete labeling of the Cre-loxP system. Under our experimental condition (with one dose of tamoxifen at low concentration), Cre does not completely induce genetic recombination. Therefore, a subset of PDGFRα+ expressing cells are not efficiently labeled with RFP+. We have added this discussion (Discussion, fourth paragraph). The reason that we prefer to use lower dose of tamoxifen is because there are associated drawbacks using high doses of tamoxifen reported by Ye et al., 2015, (included in the second paragraph of the subsection “Developmental adipocytes derive from a PDGFRα+ cell source”), which may confound our current lineage tracing studies. We have used the method for many other inducible drivers and found that they are able to generate recombination in majority of the expected cells.

Also, a paper cited by the authors uses a MerCreMer approach for inducible labelling of PDGFRα expressing cells (Cattaneo et al., 2020) and the results contradict the ones in this manuscript. May all be a problem with the PDGFRα-CRE^ERT2^ model? May it not get enough Cre activity when the mice are old by the normal tamoxifen treatments? The authors should comment more in their Discussion about these caveats and if possible, directly compare young and adult APC in these regards.

We agree with the reviewer and have discussed this discrepancy in detail in our revised manuscript. We believe that this discrepancy is likely due to multiple factors. Cre activity at different ages may contribute to the labeling efficiency and therefore lead to different lineage tracing. In addition, as pointed out in this new study Cattaneo et al., 2020, Pdgfrα-MerCreMer was produced by knocking in the inducible Cre cassette into the PDGFRα locus, which may closely represent endogenous expression but also disrupted the endogenous transcript, which may confound the findings (Ding et al., 2013). Our Pdgfrα-Cre^ERT2^ was generated using BAC transgenic mice, which did not affect endogenous expression but may not fully recapitulate the endogenous Pdgfrα expression. Therefore, we had carefully characterized the Pdgfrα-Cre^ERT2^ model in our previously paper Jiang et al., 2017a, (Figure 3A-E). We observed that PDGFRα-RFP+ cells were 100% positive for PDGFRα antibody staining, indicating that a high correspondence between reporter and endogenous Pdgfrα expression. In addition, we found that the recombination efficiency was high (70%). The only difference compared to this study was that we reduced the tamoxifen injection just for one day instead of 2 days. In this new version, we have expanded our discussion about these different systems used (Discussion, fourth paragraph).

2) The authors suggest that PDGFRα signaling have no function in adult APC. The results fall short for this affirmation as the authors already point out in the Discussion because we cannot be sure of this unless a proadipogenic and a fibrogenic challenge is done in the mice and fibrosis is actually assessed. I consider that shifting their conclusion and figure legend title to directly reflect their very interesting results would be a good idea.

We agree with the reviewers that our data show that PDGFRα signaling specifically in SMA+ cells does not play a significant role in SMA+ progenitors’ differentiation into adipocytes under physiological condition. It is unclear at this point whether PDGFRα signaling in adult APCs has other important functions under pathological conditions. For example, PDGFRα signaling might be essential for the obesogenic expansion and fibrotic response of WAT induced by HFD. This hypothesis, while speculative, is consistent with our developmental fate-mapping data under HFD condition. Further study is required to address the question of obesogenic expansion cellular source and regulatory mechanisms. Therefore, we would like to soft-pedal some of these statements and have shifted our conclusions (Abstract), text (Introduction, last paragraph, subsection “Developmental and adult PDGFRα+ cells have distinct molecular and functional signatures”, last paragraph) and figure legend of (Figure 4 and Figure 5), and discussed this in detail in the revised manuscript Discussion section.

Revisions expected in follow-up work:3) Do PPARγtTA; TRE-Cre; TRE-H2B-GFP; PDGFRα^fl/fl^ mice recover normal fat mass by P120 as the model put forward suggests?

We agree with the reviewers that this is an important question to ask, which we have not addressed in the manuscript. If these mice recover normal fat mass by P120, just like the Pdgfrα-Cre; Pparγ^f/f^, the data will further support our model. However, if we do not see the mice regain fat mass, we still cannot rule out our model. This is because adult APCs may need the precise environment to function, such as the extant structure and establishment of developmental depot. One way to circumvent this limitation is to use the Dox-off system, Dox will be added before conception and throughout embryogenesis until P60. That is; Pdgfrα will be conditionally deleted in adult intact WAT at P60 therefore avoiding developmental WAT defects. In the future studies, we will perform both experiments: 1) without Dox treatment, analyze the mice at P120; 2) with Dox on until P60 and then off Dox from P60-P120, then analyze the mice at P120.

4) The experiment in Figure 3C is definitive. Do the authors want to mention why there is not a more thorough analysis of these mice?

We agree with the reviewers that our Pdgfrα-Cre; Pparγ^f/f^ experiments were only focused on adipose tissue size and morphology. And we observed these mice had a severe loss of WAT tissue at P60 but regained WAT size when we analyzed the tissues at P120. However, these mice were recently generated and we have not fully characterized other phenotypes of these mice in detail. Therefore, we would remove the data from this manuscript if the reviewers agree. We will expand these mice and further analyze their adiposity and metabolic phenotypes together with our Pdgfrα-Cre^ERT2^-DTA (cell ablating model) in the near future when we have more litters and the Covid-19 pandemic subdues.

5) Figure 5 needs additional analysis of the browning status of the available tissues to fulfil the conclusion. As of now we don't know if both genotypes have same levels of UCP1 in white and brown fats.

We have added a real-time qPCR analysis of Ucp1 and a few of other thermogenic gene expressions in Figure 5—figure supplement 1B and text (subsection “PDGFRα in adult SMA+ APCs is not required for adult white or beige adipogenesis under physiological conditions”).